# Biologically-Based Computation: How Neural Details and Dynamics Are Suited for Implementing a Variety of Algorithms

**DOI:** 10.3390/brainsci13020245

**Published:** 2023-01-31

**Authors:** Nicole Sandra-Yaffa Dumont, Andreas Stöckel, P. Michael Furlong, Madeleine Bartlett, Chris Eliasmith, Terrence C. Stewart

**Affiliations:** 1Centre for Theoretical Neuroscience, University of Waterloo, Waterloo, ON N2L 3G1, Canada; 2Applied Brain Research Inc., Waterloo, ON N2T 1G9, Canada; 3National Research Council, University of Waterloo Collaboration Centre, Waterloo, ON N2L 3G1, Canada

**Keywords:** neural engineering framework, spatiotemporal representation, cognitive modelling, spatial semantic pointers, time cells

## Abstract

The Neural Engineering Framework (Eliasmith & Anderson, 2003) is a long-standing method for implementing high-level algorithms constrained by low-level neurobiological details. In recent years, this method has been expanded to incorporate more biological details and applied to new tasks. This paper brings together these ongoing research strands, presenting them in a common framework. We expand on the NEF’s core principles of (a) specifying the desired tuning curves of neurons in different parts of the model, (b) defining the computational relationships between the values represented by the neurons in different parts of the model, and (c) finding the synaptic connection weights that will cause those computations and tuning curves. In particular, we show how to extend this to include complex spatiotemporal tuning curves, and then apply this approach to produce functional computational models of grid cells, time cells, path integration, sparse representations, probabilistic representations, and symbolic representations in the brain.

## 1. Introduction

Over the past two decades, the Neural Engineering Framework (NEF; [1]) has been used to build a wide variety of functional, biologically plausible neural networks, including what remains the world’s largest functional brain model, Spaun [2,3]. Over that 20-year period, some of the core theoretical assumptions of the NEF have remained constant, while others have evolved and been significantly extended. As well, a variety of specific representations and neural algorithms have been proposed and implemented using the NEF methods. In this paper, we provide a summary of some of this work, focusing on results that arise from expanding the NEF’s principles while adopting its core assumptions regarding continuous spatiotemporal neural representations as being the best way to characterize neurobiological systems.

We begin by summarizing a general theoretical approach to spatiotemporal modeling that extends the applicability of the original NEF principles to a wider variety of neural dynamics. Specifically, the new techniques allow for capturing more sophisticated single neuron dynamics (e.g., adaptation) and more complex synaptic dynamics (e.g., alpha synapses) while retaining the NEF’s ability to model arbitrary nonlinear dynamical systems at the network level. Critically, as well, this NEF extension uses continuous spatiotemporal representations—thus it “starts with continuity” in the same manner as the original NEF.

We believe that embracing continuity provides an important distinction from most methods for building neural networks. Indeed, the vast majority of the largest and most useful neural networks assume discretization in state and time (see, e.g., the new Gato system [4] or AlphaGo from DeepMind [5]). While the theoretical differences between continuous and discrete representations are subtle, since any continuous representation can be approximated to an arbitrary degree of precision by a discrete one, we describe specific algorithms and representations that are derived in a continuous setting, and outperform standard methods even after they are discretized. That is, the NEF-style approach of “continuity first, discretize later,” has led to improvements in state-of-the-art neural networks [6,7].

However, our focus here is not on benchmarking neural networks on machine learning tasks, but rather on methods for biological modeling. It should be noted that here, too, the majority of methods for building models of cognitive systems—which we consider “scaled up” biological modeling—are discrete in time and space. This includes both more traditional approaches, like ACT-R [8] and Soar [9], and many neural network approaches that assume localist representations, or do not define the continuous dynamics of individual neurons [10]. In contrast, NEF-style approaches provide a foundation for continuous methods of modeling cognition, like the Semantic Pointer Architecture (SPA; [11]). We believe the SPA and NEF are complementary, in that the SPA attempts to address *what* brains compute, where as the NEF attempts to address *how* brains compute. To this end, the NEF has been used to implement the SPA-derived Spaun model, which consists of over 6 million neurons and 20 billion connections, and is able to perform a wide variety of motor, perceptual, and cognitive tasks [2,3].

We note this connection between NEF methods and cognitive modeling to provide a broader context for the work we discuss in this paper. Here, we focus on describing several specific examples of using continuous spatiotemporal representations to account for neural processing of space and time. While we do not provide large-scale cognitive examples, the methods we show here set the stage for inclusion into an improved Spaun-like model, as Spaun currently does not yet include clear examples of processing continuous time and space at the cognitive level (e.g., performing prospective and retrospective interval estimation, navigation through large environments, and so on). As such, the present work is best seen as a collection of examples that are stepping stones towards a larger combined model of how continuous time and space algorithms can perform biologically relevant tasks, while capturing central aspects of neural data in a consistent manner.

The primary benefit of our approach is that we have developed a generalized framework for reasoning about continuous variables in neural and cognitive models which both conform to neural data and explain functional behavior. In this way we close the explanatory gap between models and biological systems for a wide range of behaviors. Specifically, through spatiotemporal tuning curves we increase the biological fidelity of neural models, allowing us to design neural populations that have desired characteristics. Algorithmically speaking, this family of methods has led us to identify the Legendre Delay Network (LDN) and the Legendre Memory Unit (LMU) [6], a recurrent neural network which reproduces time cell data and provides a memory function that allows for the optimal storage of temporal events, without reliance on a particular discretization of time. In the spatial domain, we exploit Spatial Semantic Pointer (SSP) representations of continuous state spaces that can reproduce the firing patterns of grid cells [12] which we then use to develop models of path integration and cognitive mapping. Thus, our models of brain function are spatiotemporally continuous from top to bottom.

In addition, the use of continuous models to improve biological fidelity has benefits beyond modeling the brain. We have also found that they can improve the efficiency and performance of algorithms for standard machine learning, including Reinforcement Learning (RL). For instance, the LMU shows improvement over transformer models [7], and the SSP outperforms other continuous encoding methods on the majority of 122 Machine Learning (ML) benchmarks [13], and can reduce the number of trials required to learn in RL applications [14].

Overall, the unifying theme that runs through this work is the use of continuous spatiotemporal representations and dynamics in our models. We hypothesize that these representations and dynamics increase the performance of algorithms for AI, ML, and RL, while also increasing the biological fidelity of neural models.

To summarize, this work presents extensions to the NEF and a collection of examples that demonstrate the utility of our approach. We begin with a review of foundational methodology, present a collection of computational models grouped by topic, and conclude with a discussion that ties our findings together. Starting with Section 2, *A general approach to spatiotemporal modelling*, we describe the methods of the NEF. The standard NEF principles we adopt are reviewed in Section 2.1 and Section 2.2, while Section 2.3, Section 2.4 and Section 2.5 present our novel extensions to the NEF. After introducing our general approach, we show how these methods can be used in an array of tasks. In Section 3, *Time*, we introduce a series of examples of modeling temporal representation and phenomena using our approach. We present a model capturing the firing of “time cells” in the hippocampus. We then use similar algorithmic methods to develop an online learning system that makes multi-step predictions of a time-varying signal, providing a suggestion for how biological organisms might learn to predict arbitrary temporal sequences. Next, in Section 4, *Space*, we turn to methods for processing continuous space. Specifically, we describe a new kind of neural representation that captures a variety of cellular responses, including grid, place, object vector, and border cells. We then use that same representation to demonstrate how sparsity impacts the performance of a simple reinforcement learning network. In Section 5, *Space and time*, we combine continuous space and time in a path integrator network. We not only show how such a network can perform path integration, but we extend it to perform temporal integration, representing its recent path explicitly at any moment in time. In Section 6, *Quasi-probability representations*, we turn to a more recent development that extends our continuous spatial representation technique to apply to representation of probabilities. We show that these methods provide for natural ways to implement standard probabilistic algorithms efficiently in a neural substrate. Finally, in Section 7, *Discussion*, we integrate these diverse research topics, draw lessons about the value of spatiotemporal continuity in neural modeling, and consider future applications of this work.

## 2. A General Approach to Spatiotemporal Modelling

A central assumption of the NEF is that neural computation relies on neurons being *spatiotemporally tuned*. In other words, we assume that there is a systematic relationship between the stimulus history of a neuron and its present activity. This rather mild assumption gives rise to a method that allows us to construct large-scale models of neurobiological systems by compositing together smaller models.

Specifically, rather than performing end-to-end training of a complete network for a particular task, we break that task into smaller parts, train networks for each part, and combine them back together. Since each of these parts is simpler than the whole, their training can be faster. (Though, it should be noted that end-to-end optimization on the final network can still be performed.)The trade-off is that we are imposing our own structure and theoretical hypotheses onto the model. If these theoretical hypotheses as to how a task is divided among components are sub-optimal, then the resulting model will also be sub-optimal. Indeed, it is for this reason that we can claim that the models so constructed are also tests of underlying theories as to how biological neural systems (and artificial neural machine learning systems) break down tasks into individual computations.

Generally speaking, our technique can be divided into three steps, capturing and expanding on the three principles of the original NEF (indicated in parentheses):1.assigning tuning curves to individual neurons (*representation*);2.defining relationships between the variables the neurons are tuned over (*computation* and *dynamics*); and3.finding synaptic weights such that both the tuning and the desired computation emerge in the final network (*implementation*, which was originally combined with the other three principles).

Importantly, the first two steps merely describe a *normative constraint*, that is, we wish for our network to behave in a certain way. Once we have solved for synaptic weights, these normative constraints are no longer used and the network can be fully reduced to spiking neurons connected through synaptic weights and filters. The final synaptic connection weights and neural dynamics are responsible for causing the desired tuning curves to arise.

### 2.1. Tuning Curves as Building Blocks for Models of Neurobiological Systems

To explain the first step of our technique, namely assigning desired tuning curves to individual neurons, we first need to define our notion of spatiotemporal tuning more precisely. Specifically, we define “spatiotemporal tuning over *x*” as a systematic relationship between the stimulus history x(t) and the activity $a_{i}\left(x\right)$ of a neuron *i*. In everything that follows, the neural activity $a_{i}\left(x\right)$ can be either spiking or non-spiking, although we will generally write the non-spiking version for simplicity of notation. By convention, we define x(0) as the present stimulus. For t<0 the stimulus x(t) encodes past stimuli; the stimulus is undefined for t>0 to ensure causality. We call the mapping from *x* onto $a_{i}\left(x\right)$ a “spatiotemporal tuning curve”.

This definition may seem rather abstract. To provide some intuition, first consider *atemporal* tuning curves $a_{i}\left(x\right)$, that is, mappings between static *d*-dimensional vectors $x\in X\subset R^{d}$, and momentary activities $a_{i}$. This was the standard case in the original formulation of the NEF. Typically, the variable x over which a tuning curve is defined describes some stimulus presented to an experimental animal. For example, in the seminal Hubel and Wiesel experiment [15], x is the orientation φ of a bar of light, whereas $a_{i}\left(x\right)$ could be defined as the average momentary firing rate during stimulus presentation (Figure 1A,B).

However, since we defined these atemporal tuning curves as mappings over some *d*-dimensional vector space, we do not have to limit tuning curves to scalar variables such as φ. Instead, tuning curves can be defined over various modalities and mathematical objects, such as images, sounds, probability densities and even symbols [1,11,16]: anything that can be treated as a *d*-dimensional vector. Keeping this in mind, and considering the Hubel and Wiesel experiment mentioned above, we can make three key observations about tuning curves:

*1. There is flexibility in modeling the stimulus.* As mentioned above, an intuitive choice for modelling the stimulus variable x would be to use the scalar $\varphi \in [0^{\circ },180^{\circ })$. Perhaps less intuitively, but equivalently, we could use a vector x=(cos(2φ),sin(2φ)). As we discuss below, this particular choice simplifies modelling bell-shaped tuning curves. More generally speaking, observe that the choice of the variable x over which we define $a_{i}\left(x\right)$ is arbitrary; yet some x simplify mathematical modelling of certain tuning curves observed in neurobiological systems. Further examples of this principle include Spatial Semantic Pointers (SSPs) for modelling place and grid cell tuning, and the Legendre Delay Network (LDN) for modelling time cell tuning; we discuss these in more detail in Section 3 and Section 4.

*2. Tuning curves describe network-level phenomena.* A tuning curve summarises the properties of the entire neural circuitry connecting from the stimulus source to the characterised neuron. For example, the neurons in visual cortex analysed by Hubel and Wiesel are not directly connected to the sensory organs; instead, the retinal image has been transformed by various intermediate neurons. The tuning curve is thus not just the property of an individual neuron, but an emergent phenomenon of the *entire* neural system. The challenge in building a computational model of this system is thus to find the connection weights that will cause these tuning curves to appear, given the particular neural dynamics (synapses, spikes, etc.) that are used in the model. This is what the NEF is designed to do.

*3. Tuning curves can be defined over internal representations.* A direct consequence of the above is that x does not necessarily have to be a well-identifiable *external* stimulus. Instead, we can model tuning curves $a_{i}\left(x\right)$ over any quantity x that we think may be represented by neurons that directly or indirectly project onto our target neuron *i*. Examples for such x include internally generated motor commands (e.g., the desired walking speed) up to vector-representations of abstract concepts in prefrontal cortex [11,16].

### 2.2. A Linear-Nonlinear Atemporal Tuning Curve Model

While the techniques presented here work with any tuning curve shape, we typically use a linear-nonlinear tuning curve model. Using this kind of model can drastically reduce the required computational resources for training and simulation [17], but it is not required for the NEF techniques to work. We define an atemporal linear-nonlinear tuning curve as
(1)$$\begin{array}{cc} a_{i}\left(x\right) & =G\left[J_{i}\left(x\right)\right]=G\left[\alpha_{i}\langle e_{i},x\rangle +\beta_{i}\right]\;. \end{array}$$

Here, *G* is a rate-approximation of the spiking neuron model used in the network. This nonlinear function maps average input currents *J* onto average momentary firing rates. We typically use the spiking Leaky Integrate-and-Fire (LIF) neuron model, for which a closed-form expression for the momentary firing rate *G* exists; however, the same methods can be used with non-spiking neurons (e.g., ReLU, Sigmoid), single-compartment spiking neurons (e.g., LIF, ALIF, Izhikevich) and multi-compartment neuron models as well [1,18,19]. For more complex neuron models *G* must be estimated using iterative numerical methods. As a reminder, while we use rate approximations to design tuning curves, at simulation time we use the actual neuron model.

Continuing with our description of Equation (Equation 1), we use a current-translation function J(x) to map stimuli x onto a somatic current. Within this function, the linear operator 〈·,·〉 denotes an inner product, while the unit vector $e_{i}$ is the *preferred direction* or *encoder* of the neuron, $\alpha_{i}$ is a scaling factor, and $\beta_{i}$ is a constant bias current. When modeling a neurobiological system, $e_{i}$, $\alpha_{i}$, $\beta_{i}$ can be sampled such that the resulting tuning curves match key properties of the modeled neurons, such as maximum firing rates and the number of active neurons for a particular stimulus x [1].

Importantly, this tuning curve family is more powerful than it may seem. Remember from our discussion above that the exact representation of a stimulus as a vector x is an arbitrary modeling choice; we can, for example, make the dimensionality *d* as large as we would like. (The number of dimensions *d* is limited in practice. Decoding x from a group of neurons requires diverse neural tuning that covers all different “directions” in the *d*-dimensional space. This number of directions grows exponentially in *d*, unless the vector space X has special structure.). In turn, this allows us to model any tuning curve shape by representing x in a high-dimensional space and projecting onto a current *J* by choosing an appropriate encoder $e_{i}$. This “projection trick” is a common technique borrowed from machine learning, and can, for some stimulus mappings, be interpreted as an instance of the “kernel trick” [20]; we revisit this in the context of Spatial Semantic Pointers (SSPs) below. Examples of nonlinear-linear tuning curves are depicted in Figure 2, which depicts the tuning curves of various neurons used in the models discussed in this article.

### 2.3. Spatiotemporal Tuning Curves

We now introduce *time* back into our models. That is, we extend the linear-nonlinear tuning curve model from Equation (Equation 1) so that it supports spatiotemporal tuning. In other words, we can define $a_{i}$ as a pattern of activity over time as a response to the temporal *signalsx* instead of the atemporal vectors x. This can be desirable, since the response of individual neurons in the brain depends on the stimulus history *x* and not only on the momentary stimulus x. Even the original data from the Hubel and Wiesel experiment (Figure 1B) clearly shows a strong decay in activity after the stimulus onset.

To account for this in our tuning curve model, we need to replace the atemporal stimulus x with the past stimulus history $x:R^{-}⟶X\subset R^{d}$, and the encoder (i.e., the “preferred direction”) $e_{i}$ with a temporal encoder $e_{i}:R^{+}⟶X$. Then, the linear-nonlinear tuning curve model returning the current activity of the *i*th neuron in the network is:(2)$$\begin{array}{cc} a_{i}\left(x\right) & =G\left[J_{i}\left(x\right)\right]=G\left[\alpha_{i}\int_{-\infty }^{0}\langle e_{i}\left(-\tau \right),x\left(\tau \right)\rangle \;d\tau +\beta_{i}\right]\;. \end{array}$$

Assuming that $e_{i}$ is normalised, the integral term in this equation can be thought of as computing a cosine similarity between the actual stimulus history *x* and some “preferred” stimulus history $e_{i}$. That is, each neuron has a preferred pattern *over time* that will cause that neuron to fire maximally. This is the temporal generalization of the more standard preferred stimuli used in Figure 2. Similar models of spatiotemporal tuning have been explored since the 1980s in the context of motion perception [21,22,23]. For example, consider a neuron in visual cortex tuned to downwards motion. As is depicted in Figure 3, we can model such a neuron by setting the temporal encoder $e_{i}\left(t\right)$ to a prototypical downwards-moving pattern.

While we can specify anything we want as the spatiotemporal tuning curve, we can also define them on theoretical grounds. For example, we can model the rapid decay in activity in the Hubel-and-Wiesel experiment (Figure 1B) by setting $e_{i}$ to the impulse response of a high-pass filter combined with the original atemporal $e_{i}$ that picked out the particular direction φ to which the neuron was sensitive.

Note that it is possible to partially account for the temporal properties of the model neuron by replacing *G* in Equation (Equation 2) with a temporal response model Ω[J]. This can be used to account for neural adaptation [24]. Similar formalizations have been used to extend the NEF to support a variety of biologically detailed neurons [19].

Of course, for all of the above it must be noted that all we are doing is specifying the desired behavior of the neurons. We are using the temporal encoders $e_{i}$ to indicate the behavior we want from the model, but we have not yet determined how these responses actually arise from the connectivity of the network. Importantly, this tuning curve formulation is agnostic as to whether the spatiotemporal behaviour stems from some intrinsic neural dynamics, is realised on a network level by combining pre-synaptic activity, results from a specific synaptic filter, or some combination of these factors. All we have done so far is indicated the desired behavior and the last step in the NEF process will take care of determining how this arises.

### 2.4. Transforming Represented Values

The second step of our modeling technique is to decide what computation should be performed by the network. Put differently, and assuming that pre-synaptic neurons are tuned to signals $x=(x_{1},\cdots ,x_{d})$, we need to specify how these signals influence the activity of the neurons in the post-population. Mathematically, we can establish this relationship by specifying a function y=f(x) that defines how the signals the pre-populations are tuned to relate to the signal *y* to which the post-neurons are tuned. Once we have defined these relationships, we can solve for synaptic weights that establish these relationships.

As an atemporal example, consider a population of neurons tuned to a scalar *x*. We can then express that a target population should be tuned to a scalar *y* that happens to be the square of *x*, i.e., $y=f\left(x\right)=x^{2}$ (Figure 4). Of course, we can also define multivariate mappings. For example, if a pre-population is tuned to $x=(x_{1},x_{2})$, then we can define that $y=f\left(x\right)=x_{1}x_{2}$.

This example can even be extended to the time dimension; for example, we can state that the target population should represent the product between $x_{1}$ and $x_{2}$ as it was $\theta_{1}$ and $\theta_{2}$ seconds ago, that is $f\left(x_{1},x_{2}\right)\left(t\right)=x_{1}\left(t-\theta_{1}\right)\;\cdot \;x_{2}\left(t-\theta_{2}\right)$. We can similarly construct dynamical systems such as $\dot{x}=Ax+Bu$ or even $\dot{x}=f\left(x,u\right)$. This formulation is useful for defining integrators (which can act as memories, since they maintain their value when given no input), oscillators (useful for pattern generators such as walking or swimming motions), and other dynamic systems.

Importantly, as with the original NEF, while it is often convenient to define these mappings *f* in terms of a mathematical function, the same methods work in cases where we only have samples $x_{1}\mapsto y_{1},\ldots ,x_{N}\mapsto y_{N}$ as well. Such samples could for example stem from empirical studies, or be the result of sampling a computer program.

Of course, merely postulating that the neural network should perform a certain computation does not guarantee that it is actually possible to perform this computation in the final network. Indeed, the network will in general only approximate the desired function, and if it lacks the correct neural resources, it may be a poor approximation. For example, in the case of synapses that linearly combine the pre-synaptic activities (that is, a current-based synapse model [27]), a function can only be well-approximated by the network if it is also possible to linearly combine the input neuron’s filtered spatiotemporal tuning curves to approximate that desired function. That is, knowing the spatiotemporal tuning curves and synaptic filters puts constraints on the class of functions that can be computed by neural connections coming out of those neurons. For example, it is not possible to accurately compute the product between $x_{1}$ and $x_{2}$ unless there are neurons whose $e_{i}$ values are non-zero for both $x_{1}$ and $x_{2}$ [1]. This limitation can be overcome by including passive dendritic nonlinearities. In this case, a larger class of nonlinear functions including $x_{1}\;\cdot \;x_{2}$ can be computed even if $x_{1}$ and $x_{2}$ are represented in independently tuned populations [18,28].

### 2.5. Solving for Synaptic Weights

The third and final step in our technique is to solve for synaptic weights such that the desired neural tuning and computation “emerge” in the final network. One potential approach would be to perform global optimization using gradient descent. However, globally optimizing large neural networks across time is computationally expensive and often requires a large amount of training data.

We can work around this by presuming that each neuron in our network already has the tuning that we assigned to it in step one. In a sense, we are setting up a self-fulfilling prophecy: assuming that the network behaves a certain way, we solve for the connection weights needed to make it behave that way. Specifically, for current-based synapses, solving for weights $w_{i}=\left(w_{i,1},\ldots w_{i,n}\right)$ reduces to a linear regression problem:(3)$$\begin{array}{c} w_{i}=\arg\min_{w_{i}}\;\sum_{k}\;{\left(J_{i}\left(f\left(x_{k}\right)\right)-\sum_{j}w_{ij}\left(h_{ij}*a_{j}\left(x_{k}\right)\right)\left(0\right)\right)}^{2}\;. \end{array}$$

Here, $J_{i}$ is the current that we need to inject into the *i*th post-neuron according to the tuning-curve constraint (Equation (Equation 2)). Furthermore, $h_{ij}\left(t\right)$ is the impulse response of the synaptic filter between the pre-synaptic neuron *j* and the post-synaptic neuron *i*. This term describes the post-synaptic current over time as a response to a single input spike. The term $\left(h_{ij}*a_{j}\left(x_{k}\right)\right)\left(0\right)$ describes the filtered activity of the pre-neuron in response to the represented value $x_{k}$ in the present moment, t=0.

While Equation (Equation 3) may look intimidating, solving for synaptic weights is relatively straight-forward in practice. First, we randomly sample *N* signals $x_{k}$. Given those signals, we can compute the filtered response of each pre-synaptic neuron *j* according to its tuning curve, as well as the post-synaptic current $J_{i}$ that results in the activity described by the post-synaptic tuning-curve for $f(x_{k})$. The weights $w_{i}$ are then the solution to a linear least-squares problem that approximates the desired target currents by linearly combining the pre-actvities. Note that we can also add further constraints to this equation, such as nonnegative connection weights for modeling Dale’s principle and synaptic sparsity [29], and, as mentioned above, account for passive nonlinear dendrites [18,28].

There are special cases under which Equation (Equation 3) can be simplified even further. For example, if all $h_{ij}$ are first-order exponential low-pass filters, then we can solve for weights that realise dynamical systems without the need to sample activities through time—we just need to compensate for the synaptic low-pass filter not being a perfect integrator [1,30].

Furthermore, for linear-nonlinear tuning curves, the resulting weight matrices $w_{i}$ are generally of low-rank, which can be exploited to further reduce optimization and simulation time. Indeed, the original Neural Engineering Framework was derived for these special cases [1], and the simulation tool Nengo harnesses the low-rank weight matrix to speed up simulations [17].

Importantly, note that while the weights $w_{i}$ are *locally* optimal, we are not guaranteed that the solution is *globally* optimal; this is because our solution hinges on the assumption that we can perfectly realise the desired tuning $a_{i}$ in every neuron. In practice, this is never the case, and we would have to iteratively solve Equation (Equation 3) using the *actually* achieved tuning; however, just like any other global optimization scheme, this approach does not guarantee convergence to a global optimum. Nevertheless, in practice, our general approach works well in a single iteration. In the remainder of this paper we present a variety of systems that were constructed using the technique presented here.

## 3. Time

As we discussed in the previous section, the NEF describes a method for constructing neural networks in which each neuron realises some desired spatiotemporal tuning. The goal of this section is to provide some examples that focus on the *temporal* aspect. In particular, we discuss two ways in which the techniques presented above can be used to model time cells, first by directly solving for time cell tuning and then by realising temporal bases. We close this section by demonstrating how networks realising temporal bases can be extended to predict time-series.

### 3.1. Time Cells and Temporal Bases

Time cells are neurons that exhibit a delayed reaction to a stimulus *x*. First discovered in rat hippocampus [31], time cells have been described in various brain areas, including medial prefrontal cortex, striatum, and cerebellum [32].

We can model a population of time cells using our techniques from Section 2. All we need to do is to choose appropriate temporal encoders $e_{i}$, such as the bell-shaped $e_{i}$ with different delays $\theta_{i}$ depicted in Figure 5A. In accordance with *in vivo* observations, we bias $\theta_{i}$ toward smaller values and increase the spread in activation for time-cells with larger $\theta_{i}$ [33,34]. Solving for recurrent synaptic weights using Equation (Equation 3) and simulating the network results in a recurrent spiking neural network that exhibits the desired neural impulse response. Importantly, each neuron’s activity (bottom of Figure 5A) arises from the recurrent connections with all of the other neurons, and those weights have been optimized so that each neuron’s temporal tuning curve follows the desired temporal encoder (top of Figure 5A).(Remember that the tuning curves differ from the temporal encoders depicted in Figure 5A as per Equation (Equation 2). The tuning curve additionally includes the neural nonlinearity *G* as well as gains $\alpha_{i}$ and biases $\beta_{i}$; these ingredients are what enables nonlinear computation in the first place). The resulting system does not *perfectly* exhibit the desired ideal temporal tuning, but still the overall system correctly exhibits the desired time cells. The accuracy of the match between the desired temporal tuning and the actual temporal tuning generally increases with the number of neurons.

As for why it might be useful for a biological system to have these tuning curves, remember that the set of spatiotemporal tuning curves defines the class of functions that can be decoded from these neurons. In particular, a network with time cell tuning implicitly retains a *memory* of the stimulus history over some window of time. This means that it is possible to compute functions that involve the past history of the stimulus, such as the delayed product $f\left(x_{1},x_{2}\right)\left(t\right)=x_{1}\left(t-\theta_{1}\right)\;\cdot \;x_{2}\left(t-\theta_{2}\right)$. In other words, the tuning curves of a population of time cells form a temporal basis [30,35,36].

However, this temporal basis that we created using the temporal encoders in Figure 5A is not optimal, in that it is a non-orthogonal basis [24]. Imposing dynamics onto a neuron population such that they explicitly generate an orthogonal temporal basis leads to an alternate and more principled solution to the problem of enabling computation across time. For example, and as is a recurring theme in this article, we could use a higher-dimensional representation x to achieve this complex tuning behaviour over a lower-dimensional stimulus variable *u* (cf. Section 2.1). In other words, we would like to implement a dynamical system of the form $\dot{x}=f\left(x,u\right)$ such that the impulse response of the system corresponds to an orthogonal function basis over a finite time window. Given an *f* with this property, we could implement the corresponding dynamics in a neural network using the methods discussed in Section 2.4. (Implementing an LTI system $\dot{x}=Ax+Bu$ as a transformation *f* is mathematically equivalent to setting the temporal encoders $e_{i}\left(t\right)$ to a linear combination of the impulse response of the individual state dimensions [24]. We can thus either think of the basis-generating LTI systems discussed below in terms of carefully choosing temporal tuning curves $e_{i}\left(t\right)$, or as implementing a dynamical system on top of a represented space x).

Indeed, dynamical systems that (approximately) generate orthogonal temporal bases exist. The seminal example of such a “basis-generating system” is the Legendre Delay Network (LDN) [30]. The LDN is the mathematically optimal implementation of a time-delay of length θ as a *q*-dimensional Linear Time-Invariant (LTI) system $\dot{x}=Ax+Bu$. By implementing a delay, the system compresses the last θ seconds of the input *u* into its state x. The impulse response of the LDN approximates the Legendre polynomials (thus providing its name), an orthogonal function basis. The Modified Fourier (MF) system is similar in principle, but generates the Fourier series as a function basis [24].

Curiously, as is depicted in Figure 5B and C, implementing these LTI systems in a neural population results in temporal tuning that is qualitatively similar to our “ad hoc” approach from Figure 5A; indeed, the LDN reproduces key features of hippocampal time cells [30]. That is, by creating this network that is just meant to optimally store information across a window in time, the resulting tuning curves are a close match to those seen in biological systems. This result is robust across a variety of temporal bases, with different bases having some advantages in certain conditions [24].

If the constraints of the original formalisation of the NEF [1] are met (e.g., synapses are homogeneous first-order low-pass filters), then realizing basis-generating LTI systems in neural populations is not only mathematically more principled, but also computationally efficient (cf. Section 2.5). If these constraints are not met, or manual control over individual tuning curves is required (as in Figure 5A), then modellers can always fall back to the newer generalized version of the NEF discussed in Section 2.

Given these temporal tuning curves, we can now define neural networks that compute functions based on the past history of the inputs. This could be simple mathematical functions such as a delayed temporal product $x_{1}\left(t-\theta_{1}\right)\;\cdot \;x_{2}\left(t-\theta_{2}\right)$. Alternatively, the network could compute any function that maps from a temporal sequence of inputs to given outputs. For example, this could be used for word recognition, if these inputs are sound frequencies and the outputs are the classification of word utterances. Indeed, the LDN can be thought of as generating an optimal reservoir for encoding information over time, and we do not need to train any of the recurrent weights (since they are solved for using the NEF). Because of this optimality, these networks outperform state-of-the-art methods such as LSTMs and transformers on tasks that require recognizing patterns over time [6,7,37] (see Section 3.4).

### 3.2. Predicting Dynamic Systems

In addition to using the NEF principles to directly solve for the connection weights that result in some desired output, we can also combine NEF methods with online learning rules to create neural networks that learn over time. This is particularly important for modelling biological systems that learn about their environment and how their bodies interact with it.

As an example, consider a network for predicting the current position of a simple arm model, i.e., a single-joint pendulum (cf. Figure 6A, B). We feed into the model the torque commands τ(t) being sent to the arm and a *delayed* version of the position of the arm $\varphi (t-\theta^{\prime })$. This delay reflects the fact that in biological systems there tends to be a significant (tens of milliseconds) delay in data returning from the sensory periphery. Feeding these two signals into a neuron population with time cell tuning, (Implementing time cell tuning over two variables $u=(u_{1},u_{2})$ can be accomplished by implementing two LDN LTI systems in the same neuron population; i.e., the represented state variable is $x=(x_{1},x_{2})$ [24]). We now have a neural population that encodes in its temporal tuning curves these two important variables (the command sent to the arm and the last known position of the arm) over a window in time.

The desired output of the system is the current position of the arm. To learn this, we need to apply an error-driven learning rule to the connection weights coming out of the neural population with the engineered tuning curves. We start these weights as initialize these weights as all zero, and then use a variant of the standard delta learning rule [38] known as the Prescribed Error Sensitivity rule (PES; [39]). The PES is conceptually similar to the delta rule ($\Delta \omega_{i}=-\lambda a_{i}\varepsilon$, where $a_{i}$ is the activity of the *i*th neuron and ε is the current error), but is adjusted to deal with spiking neurons and for the output and the error itself to be represented by neurons using the NEF tuning curve approach. In this case, the error is $\varepsilon \left(t\right)=\hat{\varphi }\left(t\right)-\varphi \left(t\right)$, the difference between the predicted and the actual angle.

Figure 6C shows that the system learns to predict the pendulum location, taking both the history of the pendulum angle and the input torque into account.

### 3.3. Learned Legendre Predictor: Online Multi-Step Prediction

For a more complex example, we turn to the problem of predicting the *future trajectory* of the arm, rather than its current position. This sort of prediction is needed for many approaches to motor planning, especially in biological contexts where there are significant delays. However, note that the above approach of using a standard error-driven learning rule cannot be applied here, because *the error cannot be known at the time of the prediction*. Since the prediction is about the future, we need to wait for the future to arrive before we can evaluate the error in our prediction.

To deal with this problem, we note that to compute the error, we need to remember our past predictions over some window of time. This is exactly what the LDN can do. Furthermore, in order to compute the weight updates for a delta-based learning rule, we also need the activity of the neurons in the past (i.e., when the prediction was made). This, again, can be encoded with an LDN.

Taking this approach, we developed the Learned Legendre Predictor (LLP) learning algorithm [40]. It uses the LDN approach to encode the past history of predictions and the past neural activity, and then applies error-driven learning to the weights once the true observations become available (i.e., for a prediction 0.5 s into the future, the true observation is available 0.5 s later). Importantly, rather than doing prediction for just one point in time in the future, it does it over an entire future trajectory, and learns based on the entire past history. Furthermore, since this history is linearly encoded in the Legendre space, the learning rule itself ends up being linear, and only requires information local to the synapses involved, making it an efficient and biologically plausible local learning rule.

Figure 7 demonstrates LLP learning to predict the movement of a ball bouncing in a box over 0.5 s into the future. Initially, the algorithm makes no useful prediction, but as it collects more observations it learns to accurately predict the future motion over that window of time. We assess the performance of the model by looking at the RMSE between the predicted, $\hat{x}$, and actual, x, trajectory of the ball at time *t* (Figure 7C).

For this case, while we use the LLP to learn the output of the network, the neurons themselves are defined using the same temporal representations used in the previous example, although here the inputs are the *x* and *y* location of the ball, using temporal tuning curves that store the previous 0.5 s of that position. As an extension, we also show the effects of using an alternative approach to representing spatial information (the SSP) that will be discussed in the next section.

Beyond biology, the LLP has machine learning applications as a memory-efficient algorithm for multi-step prediction. Offline, deep approaches to multi-step prediction require explicit storage of observed data. At the cost of compression loss, the LLP provides a time and memory-efficient, online algorithm.

### 3.4. Applications of Time Cell Tuning and Temporal Bases

So far we have presented the LDN in the context of modelling temporal prediction. However, we have also found that the LDN is a useful tool for representing time series in general machine learning applications; we refer to the LDN combined with a linear encoding layer and nonlinear readout as the “Legendre Memory Unit” (LMU) [6]. Chilkuri and Eliasmith [7] exploit the fact that the linear recurrence relations inherent to the LMU can be solved for, allowing for LMUs to be trained in parallel. This resulted in 200-fold improvement in training speed relative to LSTM networks, and 10-fold improvement in data efficiency relative to transformer models [37].

However, the LDN component of the LMU is not simply an algebraic hypothesis for representing time, it has also been mapped onto cerebellar circuits involved in learning precise timing in a simulated eye blink task [41]. Stöckel et al. [41] implements a model of cerebellum that builds on sparsely connected recurrent networks by implementing the specific dynamics of the LDN. The dynamics of the LDN are mapped on to the dynamics of populations of Golgi and granule cells. This approach reproduced temporal conditioning behaviour observed in animals, while matching the empirical properties of cells in the cerebellum.

In addition, the LDN has also been used as the basis for modelling how animals can estimate time intervals [42]. In that work, a spiking LDN model that produces an output after an interval of time is analyzed in terms of the variance of its outputs, and it is shown to produce both the behavioural scaling effects seen in humans (increased variance for increased intervals, but only for intervals larger than 500 ms) and the neural firing patterns seen in monkeys.

These results show that the LDN is not just a useful mathematical construct, but that it also has a explanatory power for understanding neurobiological models of learning in temporal tasks.

## 4. Space

Having described models employing temporal tuning, we will now shift focus to examples of the *spatial* aspect of spatiotemporal tuning. What is meant by “spatial" in this section is quite literal—we are interested in modelling neural representations of physical space, rather than abstract atemporal stimuli in general.

The existence of continuous spatial representations in animals is supported by psychology and biology. Spatiotemporal continuity informs how events and objects are perceived [43]. Animals can discover novel shortcuts in environments [44] and humans are hypothesized to estimate distances by mentally scanning continuous routes [45]. Additionally, humans seem to have an intuitive understanding of Euclidean geometry [46]—though, not all aspects of spatial perception align with Euclidean geometry [47].

Neurophysiological data has provided insights to the neural substrates of spatial representations. Numerous classes of neurons with distinct spatial tuning curves have been discovered. *Place cells* in the hippocampus fire at select locations in an environment [48]. *Boundary cells*, found in the medial entorhinal cortex (MEC) and parasubiculum, fire when an animal is near the boundaries of an environment [49]. Also in the MEC are *object vector cells*, with firing fields dependent on distance and direction to objects in the environment [50]. Of particular note are *grid cells*, which activate at hexagonally tiled points in an environment and are believed to provide a metric of space [51,52].

Given the tight relationship in the NEF between the neural tuning curves and the class of computations that can be performed using those neurons, we want methods for spatial representation that not only explain this rich diversity of tuning in neurobiological systems, but also allow for the computation of behaviorally meaningful outputs using those neurons. The spatially sensitive neurons in the hippocampus-entorhinal circuit are believed to be involved in many cognitive processes—from path integration to the formation of episodic memories.

### 4.1. Spatial Semantic Pointers

To introduce the Spatial Semantic Pointer (SSP) representation, let us start with the NEF default of a linear-nonlinear tuning curve model, $G\left[J\left(x\right)\right]=G\left[\alpha_{i}\langle e_{i},x\rangle +\beta_{i}\right]$. To assign the nonlinear tuning curves of place, boundary, and grid cells to neurons in our models, we use the aforementioned projection trick. That is, we take a low dimensional variable, $x\in R^{m}$ (where *m* is usually 2 or 3 in the context of spatial cognition), and project it into a high dimensional vector space by $\phi :R^{m}\to X\subset R^{d}$ (with d≫m), where we can specify tuning curves that reproduce neurobiological data. The resulting high dimensional vector ϕ(x) we call a SSP [53].

Specifically, the SSP mapping is defined by
(4)$$\phi \left(x\right)=\mathrm{IDFT}\left\{e^{iAx}\right\},$$
where IDFT is the inverse discrete Fourier transform, and $A\in R^{d\times m}$ is the encoding matrix of the representation. The low dimensional x is projected onto a set of *d* lines in $R^{m}$ space, given by the rows of the encoding matrix. These scalars are then stored in the phases of complex exponentials (Figure 8A). This mapping was originally devised as a continuous extension of the binding operation used in holographic reduced representations, a computational method that uses distributed representations to encode complex, structured information. (By design SSPs were developed as a type of “fractional binding”. Interestingly the resulting mapping is similar to the encoding used in Random Fourier Features (RFF), a popular kernel method in machine learning [54]).

The SSP projection requires an encoding matrix *A*. This matrix must have certain properties, such as conjugate symmetry and a row of zeros corresponding to a zero-frequency term to ensure the final SSP ϕ(x) is real-valued and unit length. Further constraints on the encoding matrix can be enforced to design tuning curves matching those of biological neurons. It should be noted the underlying mathematics of the SSP are the same as those in fractional power encoding [55,56]. The core difference is that here we show how these operations can be computed by an arbitrary neural network using the NEF.

SSPs were developed in the broader context of holographic reduced representations (HRRs). In HRRs, concepts and symbols are represented by high-dimensional vectors, which can be combined to represent organized, hierarchical data and encode the relationships between concepts. The cosine similarity between these symbol-like vectors measures the semantic similarity between symbols. Symbol-like vectors are combined by addition, to represent sets of symbols, and *binding* or circular convolution, for slot-filler representations.

To give a concrete example of these sorts of representations and how they can be used with SSPs, consider a simple 2D environment consisting of different objects and landmarks: a rat, two pieces of cheese, and a wall (see Figure 9A). The position of the rat, $\left(x_{1},y_{1}\right)$, can be encoded as a SSP, $\phi (x_{1},y_{1})$. This, in turn, can be bound (i.e., circularly convolved) with the vector representation of the *concept* of a rat, *R*, to obtain $R⊛\phi (x_{1},y_{1})$—this represents a rat at a particular location. Likewise, the vector representation of the concept of cheese can be bound with SSP encoding of their locations, $\left(x_{2},y_{2}\right)$ and $\left(x_{3},y_{3}\right)$, to obtain $C⊛\left(\phi \left(x_{2},y_{2}\right)+\phi \left(x_{3},y_{3}\right)\right)$. In this case, the sum of SSPs to used to represent a set of locations. The wall in the environment covers an area, which can be represented by integrating the SSP encoding over the area, $\int \int_{D}\phi \left(x,y\right)dA$. This can be bound with the vector representation of the concept of walls, *W*. All together, the complete environment can be represented by adding all these object-location vector encodings:(5)$$E=R⊛\phi \left(x_{1},y_{1}\right)+C⊛\left(\phi \left(x_{2},y_{2}\right)+\phi \left(x_{3},y_{3}\right)\right)+W⊛\int \int \phi \left(x,y\right)dA$$

This vector was constructed and “queried” for different locations with approximate unbinding. The results of this are shown in Figure 9B–D. The high-dimensional SSPs are visualized in this figure via their similarity to neighbouring points.

### 4.2. Place, Grid, and Border Cells

Grid cells are believed to provide a basis for hippocampal cognitive maps and are considered instrumental for path integration. The phase, scale, and orientation of the pattern varies among grid cells. Patterns of different phases are found scattered amongst each other, while the pattern spacing of neurons is organized from small to large scales modules along the dorso-ventral axis of the medial entorhinal cortex (MEC) [57].

A neuron given an SSP as input will have activity proportional to 〈e,ϕ(x)〉=$\langle \mathrm{DFT}\left\{e\right\},e^{iAx}\rangle$, i.e., a linear combination of plane waves. By careful selection of *A* and e, particular interference patterns can be generated. For example, the interference pattern of three plane waves with wave vectors 120${}^{\circ }$ apart is a hexagonally gridded pattern (see Figure 8B). Using this fact we can construct encoders that `pick out’ different grid patterns, and, thus, a neural population consisting of grid cells with varying grid phase, scale, and orientation that represent an SSP ϕ(x) (see [12] for details). An example grid cell from a simulated spiking network is shown in Figure 10A.

Other neuron types can be generated as well. A neuron whose encoder is the SSP representation of some location $\phi (x^{\prime })$ and given SSP input, will have place cell-like activity with a place field centred at $x^{\prime }$ (see Figure 10B). Object vector and border cells can be obtained from populations encoding SSPs representing vectors to objects or walls (see Figure 10C,D).

Biological plausibility aside, it is also the case that this SSP approach to representation outperforms other continuous encoding methods—radial basis functions, tile coding, one hot encodings, and normalized input—on 74 out of 122 ML benchmarks [13], including a mix of classification and regression problems. In particular, the SSPs constrained to create grid cell-like activity patterns were found to be the best encoding method on the benchmarks, while randomly chosen SSPs were the second-best encoding method. Furthermore, the grid-cell SSPs improved the speed of learning in a spatial reinforcement learning task [14].

### 4.3. Sparse Representations

One key observed feature of place cells and grid cells is that they have very sparse activity. That is, each neuron is only active for a small proportion of possible inputs. It is believed that this feature may be key to the hippocampus’ ability to store spatial maps for multiple environments and may improve episodic memory capacity [59]. Sparse coding is important as a general principle, and when we generate models with grid cells, we can also specify the sparsity when we set desired tuning curves. This corresponds to the bias current parameter, $\beta_{i}$, in Equation (Equation 1).

Since sparsity is believed to help when learning, we again turn to an online learning example that combines the NEF methods of creating neurons with particular tuning curves and more traditional methods of changing weights in the network based on an error signal. Here, we use the SSP approach to generate neurons that represent the spatial location and orientation for an agent in a Reinforcement Learning task. We then use a standard Advantage Actor-Critic learning rule to learn the Value (expected future reward) and Policy (which action to perform) for a basic MiniGrid task [60].

In this task an agent has to navigate to a goal location in an otherwise empty 8×8 grid-world environment. The agent’s state consists of its (x,y) coordinate location in space, and a vector describing the direction it is facing. This state information is fed into a population of non-spiking grid cell neurons (implemented using the approach described in Section 4.2).

The results are shown in Figure 11 which illustrates how changes in sparsity can drastically effect performance, with reliable and accurate performance occurring when 10% to 25% of the neurons are active at any given time, consistent with biological data [61]. Interestingly, having too many neurons active (more than 50%) is detrimental to performance, unless there are only a very small number of neurons.

## 5. Space and Time

We have introduced tools for incorporating spatial representations, such as the locations of objects and landmarks, in neural network models. However, in reality spatial relationships are not fixed in time; rather, they are dynamic and constantly changing as an animal moves through the environment. In order to accurately navigate and interact with the world, the brain must be able to integrate both spatial and temporal information, taking into account the changing positions of objects and the animal itself over time.

Many cognitive processes necessitate incorporating dynamics into spatial representations. An example is path integration, the process by which an animal’s self-position estimate is continuously updated based on motion information. This allows an animal to maintain a sense of its location and orientation within an environment even when it is not directly perceiving visual landmarks or other external cues.

Additionally, there are some tasks where it is also necessary to consider the history or trajectory of a dynamic variable. For example, an animal may need to remember the path it has taken in order to return to a specific location, or it may need to track the movements of a prey animal over time in order to predict its future location. In these cases, the brain must be able to integrate both spatial and temporal information in order to generate an appropriate response.

In this section we present models that make use of continuous spatiotemporal representations to perform path integration and temporal integration.

### 5.1. Path Integration

As an animal moves through an environment its internal estimate of self-position will change over time. This continuous updating of spatial information is known as path integration. The tools of the NEF allow us to model this computation using SSPs. Let ϕ(x(t)) be the SSP representation of an agent’s position estimate. The SSP estimate can be updated using velocity input $\dot{x}\left(t\right)$:(6)$$\begin{array}{cc} \dot{\phi }\left(x\left(t\right)\right) & =\mathrm{IDFT}\left\{iA\dot{x}\left(t\right)\right\}⊛\phi \left(x\left(t\right)\right), \end{array}$$
where ⊛ is circular convolution. Path integration with SSPs can be realized through repeated convolution, or by continuous rotations in the Fourier domain. Let $\bar{\phi }\left(t\right)$ be the SSP in the frequency domain. Consider the dynamics of its *j*th component,
(7)ddtReϕ¯(t)jImϕ¯(t)j=0−aj·x˙(t)aj·x˙(t)0Reϕ¯(t)jImϕ¯(t)j.
These are the dynamics of an oscillator in the complex plane with time-varying frequency, $a_{j}\;\cdot \;\dot{x}\left(t\right)$. This is a velocity controlled oscillator—the core component of oscillator-interference models of path integration—that is integrating linear motion in direction $a_{j}$.

These dynamics can be realized with the NEF to construct a model of path integration. A set of neural populations is used to represent the SSP in the Fourier domain, $\bar{\phi }\left(t\right)$; with population *j* encoding the real and imaginary parts of $\bar{\phi }{\left(t\right)}_{j}$, along with its time-varying frequency, $a_{j}\;\cdot \;\dot{x}\left(t\right)$. Each population is recurrently connected to itself. Equation (Equation 3) is used to solve for the synaptic weights on the recurrent connections that best approximate the dynamics of Equation (Equation 7). See [62] for a detailed description of this model. Figure 12 shows that this model can accurately track a rat’s trajectory over 60 s exploring a cylindrical room 180 cm in diameter.

### 5.2. Temporal Integration

While we have presented the SSP representation approach as being used for space, in general it can be used for any continuous values. One possibility is to use it as a method for storing the past history of a value over time. For example, instead of storing an (*x*, *y*) location, we can store a scalar *x* value at a particular point in time *t* as (x(t), *t*). This encoded in the same manner as above [63].
(8)$$\begin{array}{c} \phi \left(x\left(t\right),t\right)=\mathrm{IDFT}\left\{e^{i(Ax(t)+bt)}\right\} \end{array}$$

Time *t* is projected onto the high-dimensional X space via a encoding vector $b\in R^{d}$, just as x(t) is projected using *A*.

Furthermore, since an SSP can represent multiple points at once, we can also use the SSP approach to represent an entire trajectory.
(9)$$\begin{array}{cc} \Phi (t) & =\int_{\tau =0}^{t}\phi \left(x\left(\tau \right),t-\tau \right)d\tau \end{array}$$

The result, Φ(t), represents the past history of the spatial variable x in a high-dimensional vector space. As with all high-dimensional vector-based approaches, the representation will gradually become less accurate the more information is encoded.

In practice, we want to have a neural system continuously update this representation as time progresses. The idea here is that we want to take the current representation and shift it into the past, and then add in the new x(t) value. In this way, the system is constantly maintaining the past history of a value. In the ideal case, this stores an infinite amount of information (i.e., the entire past history of the variable x(t)), but due to the constraints of the system the representation will gradually become less accurate for data farther into the past.

This provides an alternate approach to the LDN for encoding information, but one that can also be neurally implemented using the NEF spatiotemporal tuning curves. This will result in different classes of tuning curves, and will be suitable for different types of computations. In particular, there is no explicit bound on the amount of time into the past that can be stored, unlike the LDN, which is specifically optimized for conditions where we know the time window that is of importance.

To implement this sort of memory as a neural system, we note that shifting the representation into the past is exactly the same operation as is occurring for the path integrator, except that the shift here is a constant, rather than a variable. Continuous updating of such an SSP can be accomplished by adding baseline frequencies (given by the encoding vector of the time in the SSP, b) to each oscillator. Baseline oscillations could correspond to the hippocampal theta oscillations (but these are not necessary for integrating ϕ(x(t)) alone [64]).

In order to build a neural system that encodes this sort of memory Φ(t), we construct a recurrent neural network that realizes a dynamical system using the methods in Section 2.4 The dynamics of this trajectory representation (shifting the current representation and adding the new value) are given by
(10)$$\begin{array}{cc} \dot{\Phi }\left(t\right) & =\mathrm{IDFT}\{ib\}⊛\Phi (t)+\phi (x(t)). \end{array}$$

Circular convolution of IDFT{ib} and Φ(t) is equivalent to multiplying Φ(t) by a circulant matrix constructed from the vector IDFT{ib}, and so this is an LTI system. In fact, this system has the same dynamics matrix as the Modified Fourier (MF) system used in Section 3.1.

Figure 13 shows the output of a such a neural system where the input is the current physical location (*x*, *y*), so the network is storing the past trajectory. For simplicity, we only consider storing *x* here, but the overall system generalizes to multiple dimensions. The encoded SSP Φ(t) is constantly updated by the recurrent neural network that computes Equation (Equation 10). To interpret Φ(t) at two points in time, we use shading to indicate the similarity (dot product) between Φ(t) and the ideal SSP for that location in the graph. Darker shading can be interpreted as an indication that the trajectory was at that location at that point in time.

Overall, these examples show that spatial and temporal information can be encoded in the same system, and that the NEF can be used to create neural networks that manipulate this information in biologically meaningful ways. The mathematics for all of these methods are continuous in time, and this continuity is taken into account by the NEF when being encoded into neurons. Thus, these examples show how the low-level temporal dynamics of synapses and neurons combine with neural connectivity to create overall behavior such as path integration or remembering a full trajectory.

## 6. Quasi-Probability Representations

In this section, we revisit the previous statement that the shading in Figure 13 can be interpreted as something like a probability, and show that this leads to a novel way to encode and manipulate probabilities in neurons, allowing for (approximate) Bayesian optimization. We start by reiterating that Spatial Semantic Pointers can represent points in both physical and arbitrary spaces (i.e., the variable *x* does not have to correspond to a physical position, but can be any vector). Furthermore, we note that the dot product between two SSP representations induces a measure of similarity that approximates kernels that are valid for use in kernel machines [56,65]. Consequently, SSPs provide tools for designing neural tuning curves that introduce probability into neural models.

The relationship between SSPs and kernel functions is defined
(11)$$\phi \left(x\right)\;\cdot \;\phi \left(x^{\prime }\right)=\mathrm{IDFT}\left\{e^{iAx}{\bar{e^{iAx^{\prime }}}}^{T}\right\}\approx k\left(x,x^{\prime }\right)$$
where k(·,·) is the kernel function that is induced using frequency components *A*, as in the Random Fourier Feature method of Rahimi and Recht [54]. It follows that weighted sums of SSP-encoded data are potential probability distributions, and probabilities are realized by the dot product between memory-vectors (sums of SSPs, such as the integral encoding path information in Equation (Equation 10)) and query points encoded as SSPs. Using the linear-nonlinear neural model, we can design neurons whose firing rates approximate a probability distribution [66]. Combining SSPs with Bayesian linear regression, we can construct populations of neurons that approximate Gaussian process regression [65].

Kernel density estimation then can be approximated by neurons using SSPs by simple substitution.
(12)$$P\left(X=x|D\right)=\frac{1}{n}\sum_{x_{i}\in D}k\left(x,x^{\prime }\right)\approx \phi \left(x\right)\;\cdot \;\frac{1}{n}\sum_{x_{i}\in D}\phi \left(x_{i}\right)$$

However, our commonly used methods for specifying frequency components, *A*, often induce kernels which are not strictly non-negative, e.g., the sinc function for uniformly generated *A*, or weighted products of sinc functions that produce the hexagonal grid pattern described in Section 4.2. Since these can be negative, we consider the represented values to be quasi-probabilities. Glad et al. [67] presented a conversion between quasi-probability values and true probabilities, given in Equation (Equation 13).
(13)$$P\left(X=x\right)\approx \max\left\{0,\phi \left(x\right)\;\cdot \;\frac{1}{n}\sum_{x_{i}\in D}\phi \left(x_{i}\right)-\xi \right\}$$

The value ξ is a correction to ensure that $\int_{x}P\left(X=x\right)dx=1$. This conversion has the form of a ReLU neuron, where either the sum of SSPs, or the query point ϕ(x) can play the role of the synaptic weights, and the term ξ, the neuron bias. Figure 14 shows the tuning curve of a ReLU neuron that approximates a probability distribution over an arbitrary probability distribution.

We can also use the kernels induced by dot product in combination with Bayesian linear regression to approximate Gaussian process regression over the structures that are implicitly represented in the encoding ϕ(x). This has permitted us to construct a model of Bayesian Optimization, which we used as a model of curiosity [66]. This approach draws a connection between information theoretic models of exploration and the tuning curves of biological neurons. Figure 15 shows that the performance of our Bayesian optimization model achieves statistically indistinguishable regret compared to exact implementations, but with substantial improvements in the computational efficiency of performing action selection.

Our probability formulation can be used as a component of larger neural models. Individual neurons can be interpreted as representing probabilities, and furthermore, this allows large groups of neurons to represent probability distributions. Represented distributions can be manipulated for different tasks (e.g., properties like entropy can be computed, distributions can be learned online etc.). Additionally, probabilities can be computed over complex, structured data (e.g., trajectories, or environment maps). The implication of this work is first, that cognitive models built using SSPs may be interpreted as probabilistic models, and second, that our quasi-probability model may be a hypothesis for how uncertainty arises in the brain.

## 7. Discussion

Our purpose in this paper has been to both describe a continuous spatiotemporal extension to the Neural Engineering Framework, and to provide a broad sampling of networks built using this approach, for the purposes of empirically establishing the breadth of the theory. It is clear that in doing so, we have introduced additional algorithmic (e.g., LDNs and LLPs) and representational (e.g., SSPs) considerations that play an important role in developing the final models. However, all of the work consistently adopts a continuous approach to characterizing the temporal and state space properties of each application and its neural implementation. It is thus the spatiotemporal approach we have taken that is the unifying thread throughout.

The seven examples we have provided cluster naturally into those focused on temporal processing (i.e., time cells and the LLP), those focused on spatial processing (i.e., hippocampal cell types, sparsity, probability), and those focused on combining time and space (i.e., path integration, and temporal integration). Furthermore, the LDN, which lies at the heart of our temporal processing examples and was originally developed to account for neural time processing, has turned out to beat state-of-the-art methods such as LSTMs and transformers on a variety of ML benchmarks in terms of both accuracy and efficiency [6,7,37]. While the utility of the LDN and LMU for machine learning provides a clear example of transferring what we learn from biological computation to engineered systems, we believe that it also provides a highly reusable component for building more sophisticated brain models and modelling a wide variety of brain areas. The ever increasing number of areas that exhibit time cell-like tuning [32] goes some way to supporting that suspicion.

Similarly, past work has shown that SSPs are particularly effective at representing continuous data for machine learning applications [13]. However, our focus here has been on demonstrating a range of applications of this new method for representing continuous spaces in a manner that is efficient for biologically plausible neural learning and computation. We have shown this both for spatial (i.e., hippocampal cell types) and spatiotemporal (i.e., path integration and temporal integration) applications. As well, we have demonstrated how this same representation can be used in a simple reinforcement learning task, and how changing the sparsity of such a representation at the neural level impacts performance. And, lastly, we described a more speculative connection between SSPs and probabilistic processing in the brain. We show that an SSP-based approach to Bayesian optimization is far more efficient than standard techniques. Given the close connection between SSPs, detailed neural implementations, and the SPA cognitive architecture, we hypothesize that this technique can provide a new perspective on probabilistic processing in the brain.

Taken together, we believe that these examples provide support for our claim that our approach is a general one for understanding the biological basis of neural computation. The most important advantage this extension has over the original NEF is the ability to include a much wider variety of single neuron and synaptic dynamics. That is, we have created a new tool in the tool-chest of neural cognitive modeling: the ability to specify that neurons in a particular part of a model have particular spatiotemporal tuning curves, and to solve for the connection weights that will cause this to occur, taking into account biological constraints. This provides the ability to systematically explore how ubiquitous dynamics, such as adaptation, can improve the efficiency of certain neural computations, while guaranteeing that the network-level functionality remains as desired. In addition to this core advantage, this approach can be applied to any variety of linear or nonlinear spatiotemporal processing, it can effectively generate known neural responses in functional models, and it connects directly to an established architecture for biological cognition.

There are some limitations to these methods. The probability work makes strong assumptions about where the representation comes from and how the network can be trained. The LLP algorithm smooths the signals it keeps track of, and performance is sensitive to hyper-parameters. In addition, other researchers use different, more broad, definitions of tuning curves, in which they capture any neural dynamics; for example, spatiotemporal dendritic interactions or effects of modulator neurotransmitters. This is perhaps outside the scope of our definition; the mapping between stimulus and spikes. Furthermore, to obtain non-linear tuning curves with the nonlinear-linear model used in this work, care is needed in selecting correct input representations—an issue akin to feature engineering in ML. However, the representational tools we’ve presented in our examples provide a good starting point for researchers seeking to utilize the NEF in their own models.

Much work remains to be done to take advantage of this approach. For instance, using these techniques to create or expand large cognitive models like Spaun should allow us to systematically explore the effects of single neuron and more sophisticated synaptic dynamics on high-level cognitive function. In addition, many of the examples above use algorithms that have not been integrated into such a large-scale model, including using LDNs for characterizing temporal and predictive behaviors, and using SSPs for navigation, learning, and probabilistic processing. It will be important to determine what kinds of single cell dynamics can be successfully incorporated into such models in practice and what limitations there are. The majority of examples we have explored to date have dynamics that are not as complicated as those found in sophisticated multi-compartmental single cells models. It is generally possible to include more detailed passive dendrite models in the NEF [28]. Additionally, active dendrites and conduction-based synapses have been modelled in, what’s called, the oracle-supervised Neural Engineering Framework (osNEF) [19]. While, this greater biological fidelity comes at the cost of a more computationally expensive training regime, this is necessary to model the dynamics of actual neurons. Indeed, active dendrites are necessary to capture the complex neuronal dynamics of, for example, FS basket cells in hippocampus [68] and Purkinje cells of the cerebellar cortex [69,70]. It remains to be seen if the tools presented in this work could be successfully deployed on the more biologically complex osNEF.

## 8. Conclusions

We have shown throughout this paper how expanding the principles of the NEF while continuing to adopt its core assumptions supports building a wide variety of biologically constrained neural models. In particular, developing models that start with continuous spatiotemporal representations and computations can be accomplished systemically and successfully for behaviours ranging from learning online prediction and motor control to spatial navigation and probabilistic inference. Of course, the examples provided in this paper, while disparate, are only a small sampling of the range of behaviours observed in biological systems. As such, the main aim of this paper is to canvas the general theoretical approach and encourage adoption of the core assumptions by demonstrating their utility. Given these successes, we can cautiously update the original observation of the NEF as “a ‘zeroth-order guess’ about how neural systems function,” (p. xiii, [1]). Since the expansions to the NEF described in this paper allow us to better capture the details of single-neuron and synaptic dynamics, we are now perhaps warranted to call this version of the NEF a “first-order guess” about how neural systems function.

## 9. Patents

The following patents relate to work reported in this paper: 17/049,943 (LMU), 63/087,100 (LMU), 17/895,910 (LLP), 62/820,089 (SSP), 17/520,379 (TSP).

## Figures and Tables

**Figure 1 brainsci-13-00245-f001:**
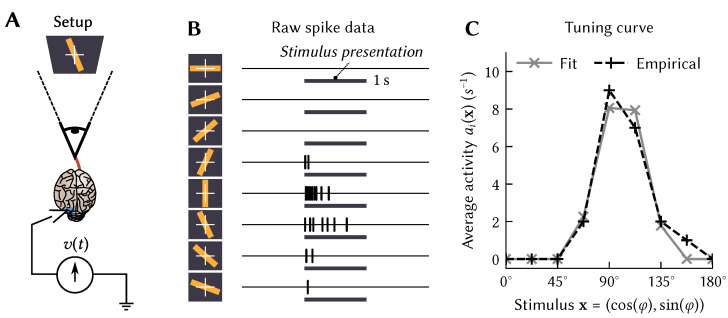
Illustration of the original Hubel and Wiesel experiment [15]. (**A**) Measuring the activity $a_{i}$ of a neuron in response to a bar of light projected onto a screen. (**B**) The neuron produces different levels of activity depending on the orientation of the bar of light x (data from [15]). (**C**) Plotting the activity $a_{i}\left(x\right)$ (dashed line) and a least-squares fit G[J(x)] (gray line; *G* is a LIF response curve).

**Figure 2 brainsci-13-00245-f002:**
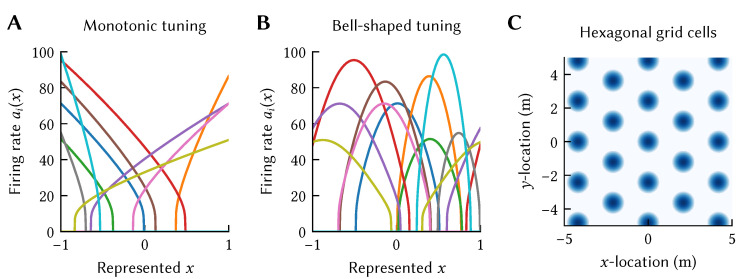
Examples of tuning curves in the linear-nonlinear tuning-curve family with a LIF nonlinearity *G*. (**A**) Using d=1 results in monotonic tuning over a quantity *x*. Coloured lines correspond to individual neurons *i*. (**B**) Bell-shaped tuning-curves can be obtained with d=2 and transforming x=(sin(πx),cos(πx)). (**C**) Tuning curve of a single hexagonal grid cell constructed using spatial semantic pointers (SSPs) with d=7 (deep blue corresponds to a firing rate of 100 Hz).

**Figure 3 brainsci-13-00245-f003:**
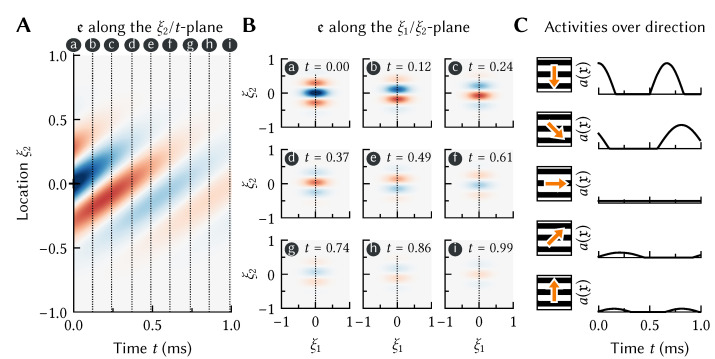
Model of a neuron in visual cortex tuned to downwards motion. The spatiotemporal encoder *e* is a two-dimensional Gabor filter sampled in two spatial dimensions $\xi_{1}$, $\xi_{2}$ that correspond to coordinates in the visual field [25,26]. (**A**,**B**) Different slices through *e*. Blue corresponds to positive, red to negative values. (**C**) Computing the neural activity for a grating pattern *x* moving in the arrow direction. Downwards motion results in the strongest response amplitude.

**Figure 4 brainsci-13-00245-f004:**
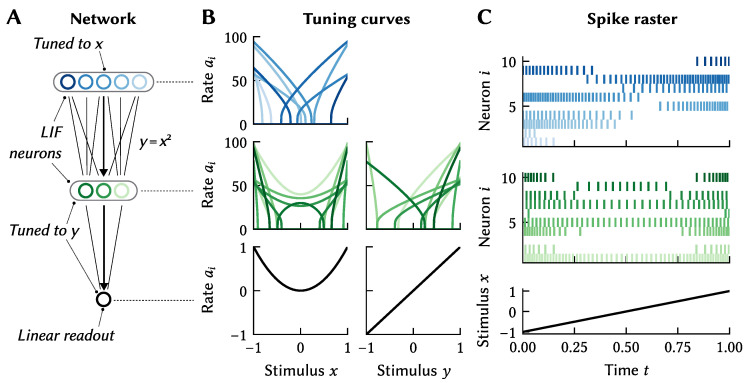
Transforming signals. (**A**) Two LIF neuron populations (blue, green) are tuned to variables *x*, *y*. The first population projects onto to the second; we impose the relationship $y=f\left(x\right)=x^{2}$ when solving for weights. The black “neuron” is a linear readout with G[J]=J. (**B**) Tuning curves of the neurons depicted in *(A)*. Left column depicts the tuning curves over *x*, right column the tuning curves over *y*. The tuning of the first (blue) population is undefined with respect to *y*. When controlling the stimulus variable *x* the network implicitly computes $f\left(x\right)=x^{2}$. (**C**) Spike raster of the two LIF populations when varying the stimulus *x* over time; although we solve for weights using a rate approximation G[J], the resulting network is compatible with spiking neurons.

**Figure 5 brainsci-13-00245-f005:**
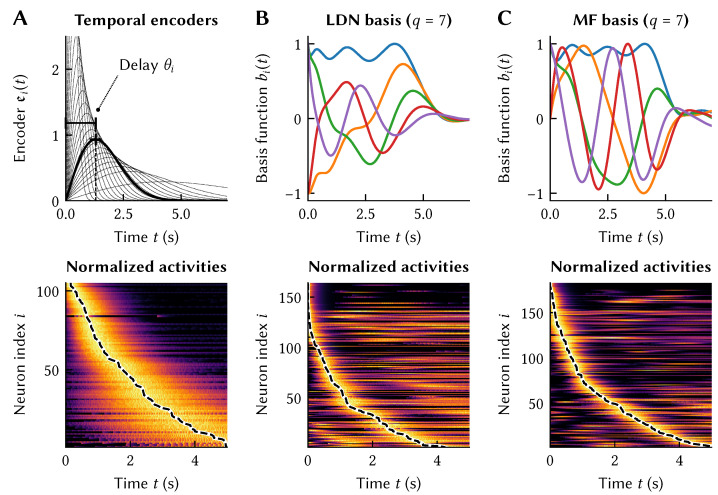
Realising time cells in NEF networks. (**A**) *Top:* Manually selected temporal encoders $e_{i}$ modelling core properties of biological time cells: bias towards shorter delays $\theta_{i}$, and larger spread in activity for larger $\theta_{i}$. *Bottom:* Activities of 200 recurrently connected integrate-and-fire neurons in response to a positive pulse after solving for weights realising the $e_{i}$. Activities are normalised to the maximum activity of each neuron (yellow). Only active neurons are depicted; 50% of the neurons are “off”-neurons that react to negative input pulses. (**B**,**C**) Qualitatively similar activities can be obtained when selecting a linear combination of temporal basis functions as temporal encoders. The basis functions depicted here are the impulse response of the Legendre Delay Network (LDN) and the Modified Fourier (MF) Linear Time Invariant (LTI) systems for q=7. Having closed-form state-space LTI systems with matrices (A,B) simplifies solving for recurrent weights in the NEF.

**Figure 6 brainsci-13-00245-f006:**
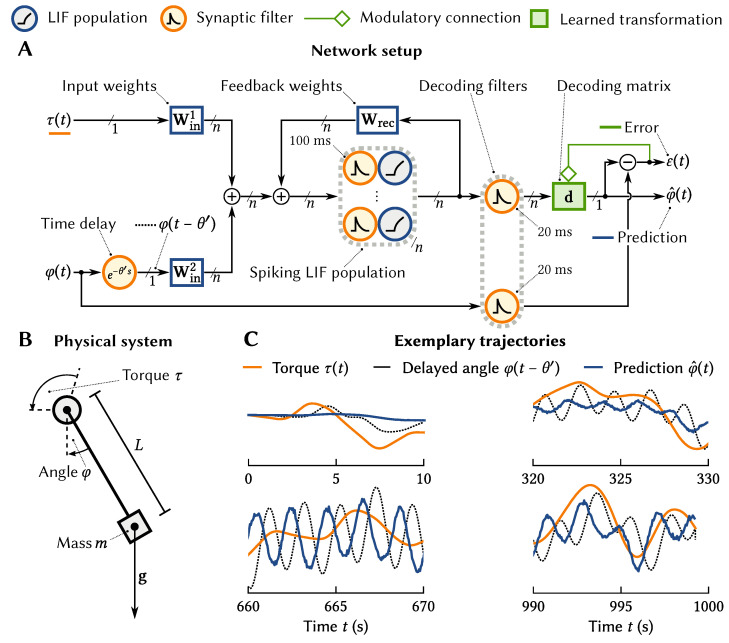
Using neurons with time cell tuning to predict nonlinear pendulum dynamics. (**A**,**B**) Overview of the experimental setup. The torque τ(t) and a delayed angle $\varphi (t-\theta^{\prime })$ are fed into a recurrent neural network with time-cell tuning over two dimensions. We use the delta learning rule to learn connection weights online that recombine the neural activities to predict the angle in $\theta^{\prime }$ seconds. (**C**) The system learns to predict the pendulum angle with a normalized RMSE of about 20%.

**Figure 7 brainsci-13-00245-f007:**
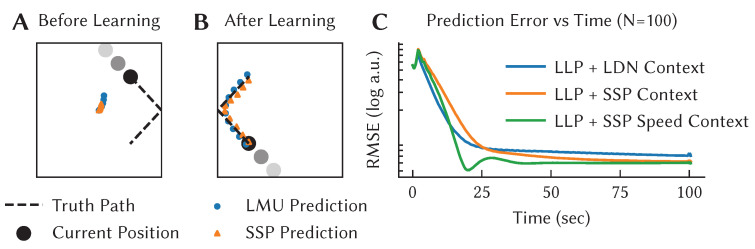
One trial of LLP learning to predict the motion of a ball bouncing off the walls of a box with lossless collisions. (**A**) Initially the system cannot predict the future motion of the ball. (**B**) Using a LDN dimensionality of q=10, the LLP learns to predict the future motion of the ball. (**C**) shows the windowed mean (window of 1 s) of the root mean square error of the predicted path for the LLP algorithm with three different context representations over 100 trials. The LDN context uses an LDN to summarize the recent motion of the ball. The SSP context encodes the current position of the ball, and the SSP Speed context encodes the position and velocity of the ball. For each context encoding we used the largest learning rate that provided a stable learning rule. The solid line is the average performance, and the shaded regions (not visible in plot) represent a 95% confidence interval. While the SSP algorithms learn more slowly than the LLP with the LDN context, they ultimately reach lower prediction error. In all cases, by working in the LDN’s compressed representation we can learn to predict delayed signals, updating historical predictions with simple linear operations.

**Figure 8 brainsci-13-00245-f008:**
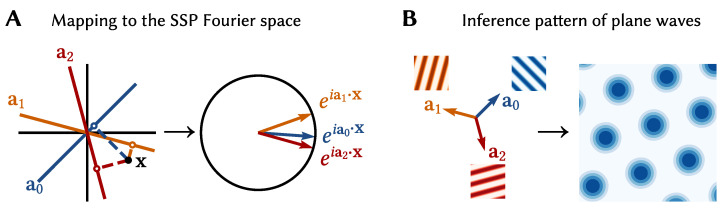
(**A**) Illustration of the projection to the frequency domain of the SSP space given in Equation (Equation 4). The dot products between a 2D variable x and a set of three vectors $\left\{a_{j}\right\}$ are cast as the phases of a set of phasors, $\left\{e^{ia_{j}\;\cdot \;x}\right\}$, which reside on the unit circle in the complex plane. The IDFT of the vector $\left[e^{ia_{0}\;\cdot \;x},e^{ia_{1}\;\cdot \;x},e^{ia_{3}\;\cdot \;x}\right]$ is a SSP representation of x, which resides in a higher dimensional vector space. In this example, the SSP is only 3-dimensional, but in practice a much larger set $\left\{a_{j}\right\}$ is used to produce high dimensional SSPs. (**B**) Consider how these phasors change for a $x^{\prime }$ traversing 2D space. The banded heat maps shows how the real part of the vectors $\left\{e^{ia_{j}\;\cdot \;x}\right\}$ repeats over a 2D region of $x^{\prime }$ values. Each component of the SSP in the Fourier domain is a plane wave with wave vector $a_{j}$. The gridded heat map is the similarity between the SSP representation of x from (A) and SSPs of neighboring points: $\phi \left(x\right)\;\cdot \;\phi \left(x^{\prime }\right)$. The similarity map is periodic due to the inference pattern of all plane waves. Here a hexgaonlly gridded similarity pattern is obtained.

**Figure 9 brainsci-13-00245-f009:**
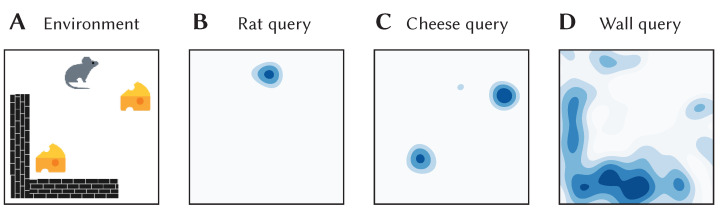
Map encoding using SSPs. (**A**) A 2D environment consisting of a rat, walls, and cheese. Information about the objects and their locations was encoded in single vector *E*, as per Equation (Equation 5). (**B**) The vector *E* was queried for the location of the rat by approximate unbinding: $E⊛R^{-1}\approx \phi \left(x_{1},y_{1}\right)+$ noise. The cosine similarity between the query output and SSP representations of points gridded over 2D space was computed and plotted to produce the above heat map. (**C**) The similarity map obtained from querying the map *E* for the location of cheese. (**D**) The similarity map obtained from querying for the wall area.

**Figure 10 brainsci-13-00245-f010:**
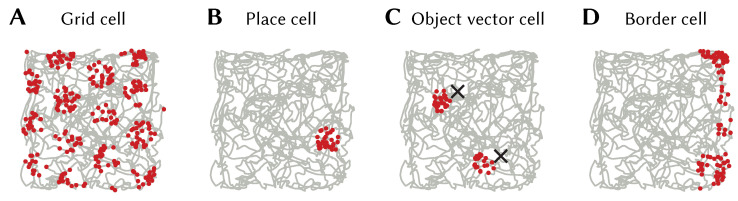
Firing patterns of LIF neurons representing SSPs. (**A**) A grid cell from a population encoding ϕ(x(t)), where x(t) is the path shown in grey (obtained from [58]). Red dots indicate the positions at which the cell fired. (**B**) A place cell from a population encoding ϕ(x(t)). (**C**) An object vector cell from a population encoding the SSP representation of the vector between x(t) and any objects in view. Object locations are marked with an ‘x’. (**D**) A border cell from a population encoding the SSP representation of the vector between x(t) and a wall along the right side of the environment.

**Figure 11 brainsci-13-00245-f011:**
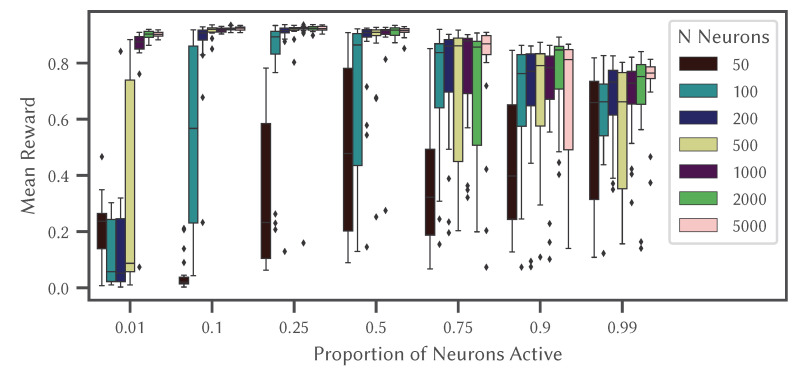
Mean reward gained over 200 learning trials for each configuration of the Actor-Critic network exploring how sparsity (proportion of neurons active at any given time) and number of neurons impacts network performance on a spatial reinforcement learning task (MiniGrid).

**Figure 12 brainsci-13-00245-f012:**
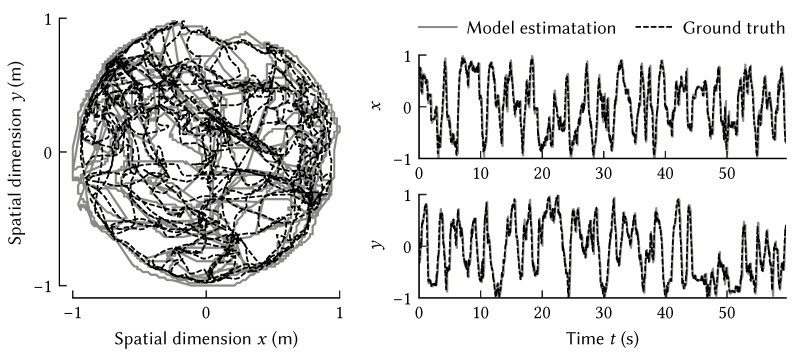
The path integration model results on a 60 s long 2D path (a rat’s trajectory running in a cylinder with a diameter of 180 cm; obtained from [51]). The grey line is the ground truth. As input the model received a initial position and the velocity along the path (computed via finite differences). The output of the model was a position estimate, in the form of an SSP, over time. The 2D path estimate plotted as a black dashed line was decoded from the raw SSP output.

**Figure 13 brainsci-13-00245-f013:**
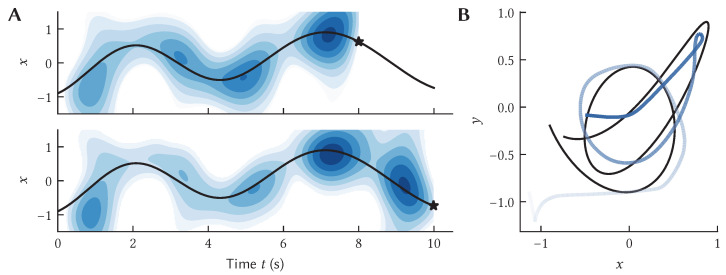
Results from temporal integration of SSPs to obtain trajectory representations. Path integration was performed on a 2D path (the black line in (**B**)). The output of the path integrator was fed into a temporal integrator, with dynamics given by Equation (Equation 10). (**A**) Two panels show the *x*-dimension of the trajectory output at different points (indicated by black stars) over the simulation time (the x axis). The output Φ(t) is visualized by a contour plot of its similarity with SSP representations across *x*-space. This is analogous to a probability distribution of the *x* position at different points in the past (see Section 6). (**B**) The 2D trajectory estimate, decoded from Φ(t) at the end of the simulation, as a blue line that fades with how far the estimate is into the past.

**Figure 14 brainsci-13-00245-f014:**
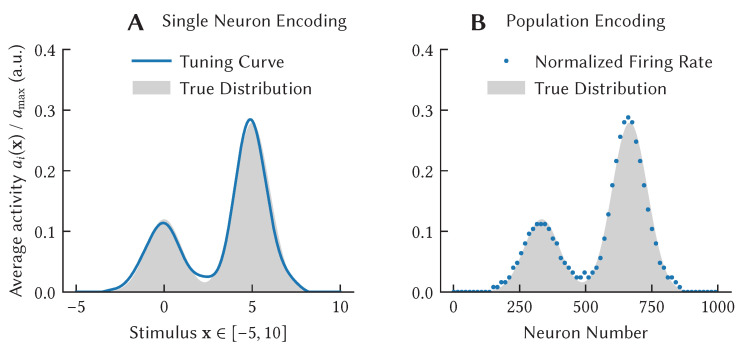
Kernel Density Estimators (KDEs; green line) approximate probability distributions (shaded region). Using the Spatial Semantic Pointer representation we can approximate the Fourier Integral Estimator (FIE)—a density estimator using a sinc kernel function. More importantly, we can represent probability with finite neural resources, and interpret operations on that representation as probability statements. Figure adapted from [66].

**Figure 15 brainsci-13-00245-f015:**
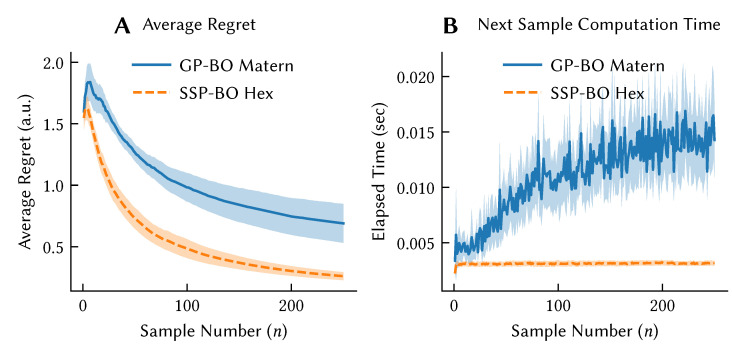
The regret performance of Bayesian optimization implemented using Gaussian processes with a Matern kernel (GP-BO Matern) and implemented using the Hexagonal Spatial Semantic Pointer representation (SSP-BO Hex) on the Himmelblau standard optimization test function (**A**). The regret performance of SSP-based algorithms is statistically equivalent to the GP methods, however, by working in the neurally-plausible feature spaces, the computation time becomes constant in the number of samples collected (**B**). Figured adapted from [65].

## Data Availability

Source code for the simulations are available at https://github.com/ctn-waterloo/brain_sciences_biological_computation_2022 (accessed on 31 December 2022).

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
