# Peer review of "Biologically-Based Computation: How Neural Details and Dynamics Are Suited for Implementing a Variety of Algorithms"

_brainsci, 2023, doi:10.3390/brainsci13020245_

Round 1

Reviewer 1 Report

Overall, the manuscript is a lot more informative but the paper is not organized properly. So, actual efforts were unable to conclude into a significant outcome. Even after reading the whole manuscript, it could not answer many questions and in the same way, making it difficult to give concrete review decisions. Please find my comments.

1.       The abstract is very short. Please also improvise the abstract. It should highlight the main contributions and achievements of the proposed work.

2.       As the paper is all about “The Neural Engineering Framework”, it should be explained in little detail, otherwise, it is difficult to get the exact context of the work.

3.       At the end of section 1, the authors should brief the paper about the paper organization. What authors have analyzed and some experimental outcomes.

4.       Authors should clarify: What is exactly known about the topic? And second What authors have newly contributed to the world through this manuscript?

Author Response

To reviewer 1,

Thank you for the feedback on our manuscript. Please see below for a point-by-point response to your comments and suggestions.

Overall, the manuscript is a lot more informative but the paper is not organized properly. So, actual efforts were unable to conclude into a significant outcome. Even after reading the whole manuscript, it could not answer many questions and in the same way, making it difficult to give concrete review decisions. Please find my comments.

We organized the paper in a somewhat unorthodox way since it is an accumulation of several threads of research expanding the NEF, rather than just a presentation on a single model or result. For this reason, rather than structure the paper in the standard introduction-methods-results-discussion fashion, we have structured it by grouping these research threads by subject: Time, Space, Space & Time, Probability (with an introduction, ‘general approach’, and discussion sections bookmarking these to tie them together). We have extended a paragraph in the introduction to explain this organization (page 3 of the revised manuscript or see response to point 3 below).

  1.The abstract is very short. Please also improvise the abstract. It should highlight the main contributions and achievements of the proposed work.

We agree that the abstract in our early version was too short. We have expanded the abstract, and placed greater emphasis on the ‘take-home’ message/ main contributions. The abstract now reads: 

The Neural Engineering Framework (Eliasmith & Anderson, 2003) is a long-standing method for implementing high-level algorithms constrained by low-level neurobiological details. In recent years, this method has been expanded to incorporate more biological details and applied to new tasks. This paper brings together these ongoing research strands, presenting them in a common framework. We expand on the NEF’s core principles of a) specifying the desired tuning curves of neurons in different parts of the model, b) defining the computational relationships between the values represented by the neurons in different parts of the model, and c) finding the synaptic connection weights that will cause those computations and tuning curves. In particular, we show how to extend this to include complex spatiotemporal tuning curves, and then apply this approach to produce functional computational models of grid cells, time cells, path integration, sparse representations, probabilistic representations, and symbolic representations in the brain.

  2. As the paper is all about “The Neural Engineering Framework”, it should be explained in little detail, otherwise, it is difficult to get the exact context of the work.

The generalized formulation of The Neural Engineering Framework (NEF) used in this work is described in Section 2 (A General Approach to Spatiotemporal Modelling). The new text added to the introduction outlining the paper organization (see above) now explains this more clearly:

Starting with section 2, A general approach to spatiotemporal modelling, we describe the methods of the NEF. The standard NEF principles we adopt are reviewed in sections 2.0 to 2.2, while sections 2.3 to 2.5 present our novel extensions to the NEF.

  3. At the end of section 1, the authors should brief the paper about the paper organization. What authors have analyzed and some experimental outcomes.

 Thank you for this suggestion of adding a paper organization brief. We have now added a few paragraphs outlining the content & main results of each section (on page 3 of the revised manuscript):

To summarize, this work presents extensions to the NEF and a collection of examples that demonstrate the utility of our approach. We begin with a review of foundational methodology, present a collection of computational models grouped by topic, and conclude with a discussion that ties our findings together.

Starting with section 2, A general approach to spatiotemporal modelling, we describe the methods of the NEF. The standard NEF principles we adopt are reviewed in sections 2.0 to 2.2, while sections 2.3 to 2.5 present our novel extensions to the NEF. After introducing our general approach, we show how these methods can be used in an array of tasks. In section 3, Time,  we introduce a series of examples of modeling temporal representation and phenomena using our approach. We present a model capturing the firing of “time cells” in the hippocampus. We then use similar algorithmic methods to develop an online learning system that makes multi-step predictions of a time-varying signal, providing a suggestion for how biological organisms might learn to predict arbitrary temporal sequences. Next, in section 4, Space, we turn to methods for processing continuous space. Specifically, we describe a new kind of neural representation that captures a variety of cellular responses, including grid, place, object vector, and border cells. We then use that same representation to demonstrate how sparsity impacts the performance of a simple reinforcement learning network. In section 5, Space and time, we combine continuous space and time in a path integrator network. We not only show how such a network can perform path integration, but we extend it to perform temporal integration, representing its recent path explicitly at any moment in time.  In section 6, Quasi-probability representations, we turn to a more recent development that extends our continuous spatial representation technique to apply to representation of probabilities. We show that these methods provide for natural ways to implement standard probabilistic algorithms efficiently in a neural substrate. Finally, in section 7, Discussion, we integrate these diverse research topics, draw lessons about the value of spatiotemporal continuity in neural modeling, and consider future applications of this work.

  4. Authors should clarify: What is exactly known about the topic? And second What authors have newly contributed to the world through this manuscript?

The main contributions of this work are the generalization of the NEF to handle spatiotemporal tuning curves, and the application of this in the array of examples presented in the paper. The new text added to the introduction, outlining section 2, clarifies this as a main contribution.

Reviewer 2 Report

The MS "Biologically-based computation: How neural details and dynamics are suited for implementing a variety of algorithms" is a high quality MS, really informative and well presented.

The quality of writing is excellent, as well as the quality of figures. For this MS I would suggest only minor changes. 

Introduction: 

Please clearly state the hypothesis of work in this section

Discussion: It would be helpful to discuss the limitations of the study, the perspective of your work.

Regarding the structure, it would be helpful to clearly separate the different sections and develop the "method" section.

Author Response

To reviewer 2,

Thank you for the feedback on our manuscript. Please see below for a point-by-point response to your comments and suggestions.

The MS "Biologically-based computation: How neural details and dynamics are suited for implementing a variety of algorithms" is a high quality MS, really informative and well presented.

Thank you!

The quality of writing is excellent, as well as the quality of figures. For this MS I would suggest only minor changes. 

Introduction: Please clearly state the hypothesis of work in this section

We have added the following text to the introduction to highlight the main contributions of this work – generalization of the NEF to handle spatiotemporal tuning curves, and the application of this in the array of examples presented in the paper – on the bottom of page 2 of the revised manuscript. Furthermore, we have modified the second-to-last paragraph of the introduction to more clearly state the hypothesis of this work:

Overall, the unifying theme that runs through this work is the use of continuous spatiotemporal representations and dynamics in our models. We hypothesize that these representations and dynamics increase the performance of algorithms for AI, ML, and RL, while also increasing the biological fidelity of neural models.

Discussion: It would be helpful to discuss the limitations of the study, the perspective of your work.

We agree with this sentiment. Certain limitations are discussed in the Discussion in the following paragraph, with the later part regarding more complex neuronal models being new to the revised versions:

Much work remains to be done to take advantage of this approach.  For instance, using these techniques to create or expand large cognitive models like Spaun should allow us to systematically explore the effects of single neuron and more sophisticated synaptic dynamics on high-level cognitive function. In addition, many of the examples above use algorithms that have not been integrated into such a large-scale model, including using LDNs for characterizing temporal and predictive behaviors, and using SSPs for navigation, learning, and probabilistic processing. It will be important to determine what kinds of single cell dynamics can be successfully incorporated into such models in practice and what limitations there are.  The majority of examples we have explored to date have dynamics that are not as complicated as those found in sophisticated multi-compartmental single cells models. It is generally possible to include more detailed passive dendrite models in the NEF [28]. Additionally, prior work has adapted the NEF to accommodate models with biologically detailed active dendrites and conduction-based synapses, but at the cost of a more computationally expensive training regime [19]. It remains to be seen if the tools presented in this work could be successfully deployed on this more biologically complex version of the NEF. 

Another paragraph has also been added (page 24) to discuss further limitations:

There are some limitations to these methods. The probability work makes strong assumptions about where the representation comes from and how the network can be trained. The LLP algorithm smooths the signals it keeps track of, and performance is sensitive to hyper-parameters. In addition, other researchers use different, more broad, definitions of tuning curves, in which they capture any neural dynamics; for example, spatiotemporal dendritic interactions or effects of modulator neurotransmitters. This is
perhaps outside the scope of our definition; the mapping between stimulus and spikes. Furthermore, to obtain non-linear tuning curves with the nonlinear-linear model used in this work, care is needed in selecting correct input representation - an issue akin to feature engineering in ML. However, the representational tools we’ve presented in our examples provide a good starting point for researchers seeking to utilize the NEF in their own models.

Regarding the structure, it would be helpful to clearly separate the different sections and develop the "method" section.

We have now expanded a paragraph (on page 3 of the revised manuscript) outlining the content & main results of each section to help clarify and differentiate the sections. This included adding text to more clearly label Section 2 (A General Approach to Spatiotemporal Modelling) as the main ‘method’:

To summarize, this work presents extensions to the NEF and a collection of examples that demonstrate the utility of our approach. We begin with a review of foundational methodology, present a collection of computational models grouped by topic, and conclude with a discussion that ties our findings together.

Starting with section 2, A general approach to spatiotemporal modelling, we describe the methods of the NEF. The standard NEF principles we adopt are reviewed in sections 2.0 to 2.2, while sections 2.3 to 2.5 present our novel extensions to the NEF. After introducing our general approach, we show how these methods can be used in an array of tasks. In section 3, Time,  we introduce a series of examples of modeling temporal representation and phenomena using our approach. We present a model capturing the firing of “time cells” in the hippocampus. We then use similar algorithmic methods to develop an online learning system that makes multi-step predictions of a time-varying signal, providing a suggestion for how biological organisms might learn to predict arbitrary temporal sequences. Next, in section 4, Space, we turn to methods for processing continuous space. Specifically, we describe a new kind of neural representation that captures a variety of cellular responses, including grid, place, object vector, and border cells. We then use that same representation to demonstrate how sparsity impacts the performance of a simple reinforcement learning network. In section 5, Space and time, we combine continuous space and time in a path integrator network. We not only show how such a network can perform path integration, but we extend it to perform temporal integration, representing its recent path explicitly at any moment in time.  In section 6, Quasi-probability representations, we turn to a more recent development that extends our continuous spatial representation technique to apply to representation of probabilities. We show that these methods provide for natural ways to implement standard probabilistic algorithms efficiently in a neural substrate. Finally, in section 7, Discussion, we integrate these diverse research topics, draw lessons about the value of spatiotemporal continuity in neural modeling, and consider future applications of this work.

Reviewer 3 Report

This is an elegant manuscript from the center for Theoretical Neuroscience at the University of Waterloo. They have made many contributions to implementing high-level algorithms constrained by biological details. This work used examples to demonstrate how continuous time and space algorithms can perform biologically relevant tasks. Overall, I have no major critiques. Here are my suggestions to improve this manuscript:

1, The authors claim to provide a general approach to spatiotemporal modeling, see tuning curves in Figure 2 and figure 4. Are these tuning curves general enough? They are probably general enough for the examples shown in this manuscript. The spatiotemporal interactions of synaptic inputs are very complicated. See the relevant examples in pyramidal neurons (Gidon & Segev, neuron 2012). The interaction is both site and timing dependent. I wonder whether this type of general relationship is available. Even if it exists, the approximation should be very complex.

2, This is still related to comment 1. In line 176, “the techniques presented here work with any tuning curve shape.” Can this work with the 2-stage nonlinear integration observed in FS basket cells (Tzilivaki et al., Nat. Comms 2019)? Also, neuronal computational properties are not static but firing state-dependent (see Phoka et al., PLos CB 2010; Zang et al., eLife 202) and regulated by modulators. It is also confusing why the authors repeatedly refer to “passive” dendrites. We all know dendrites are active, as shown in all the previously mentioned papers. Given all these factors, does the statement in lines 193-198 still hold?

The biological systems are intrinsically complex, and therefore, it is understandable that models show limitations. The authors don’t need to change their model or methods but should either tone down their claims or justify their methods through discussions.

3, the link to the code does not work.

Author Response

To reviewer 3,

Thank you for the feedback and insightful suggestions on our manuscript. Please see below for a point-by-point response to your comments.

This is an elegant manuscript from the center for Theoretical Neuroscience at the University of Waterloo. They have made many contributions to implementing high-level algorithms constrained by biological details. This work used examples to demonstrate how continuous time and space algorithms can perform biologically relevant tasks. Overall, I have no major critiques. 

Thank you!

Here are my suggestions to improve this manuscript:

1, The authors claim to provide a general approach to spatiotemporal modeling, see tuning curves in Figure 2 and figure 4. Are these tuning curves general enough? They are probably general enough for the examples shown in this manuscript. The spatiotemporal interactions of synaptic inputs are very complicated. See the relevant examples in pyramidal neurons (Gidon & Segev, neuron 2012). The interaction is both site and timing dependent. I wonder whether this type of general relationship is available. Even if it exists, the approximation should be very complex.

See response below.

2, This is still related to comment 1. In line 176, “the techniques presented here work with any tuning curve shape.” Can this work with the 2-stage nonlinear integration observed in FS basket cells (Tzilivaki et al., Nat. Comms 2019)? Also, neuronal computational properties are not static but firing state-dependent (see Phoka et al., PLos CB 2010; Zang et al., eLife 202) and regulated by modulators. It is also confusing why the authors repeatedly refer to “passive” dendrites. We all know dendrites are active, as shown in all the previously mentioned papers. Given all these factors, does the statement in lines 193-198 still hold?

The NEF as described in the paper has not been shown to support the kind of spatiotemporal dendritic interactions that you reference. The following has been added to a paragraph in the Discussion describing limitations to reference this:

In addition, other researchers use different, more broad, definitions of tuning curves, in which they capture any neural dynamics; for example, spatiotemporal dendritic interactions or effects of modulator neurotransmitters. This is perhaps outside the scope of our definition; the mapping between stimulus and spikes.

Our claim regarding “any tuning curve shape” (also see comment 2.) was solely with respect to the particular mathematical tuning curve model described by us. In other words, any tuning curve shape that fits our overall model can be used; unfortunately, the dendritic interactions referenced by you do not fit that model.

That being said, it is generally possible to include more detailed dendrite models in the NEF; however this is done at the weight solving state of the process. Particularly, the weight solver will treat the multi-channel dendrites as a computational resource to establish the desired spatiotemporal tuning according to our abstract model.

So far, we have only tested this systematically with passive dendrites (see the publications referenced in the paper). The goal of this research was to show that even modeling basic passive dendrites substantially improves the range of functions that can be computed by the NEF.

We have also adapted the NEF to work with neurons that feature more biologically detailed active dendrites (Duggins, P.; Eliasmith, C. Constructing functional models from biophysically-detailed neurons. PLoS computational biology 2022). However, as of writing, we have not demonstrated that including biologically detailed neurons is beneficial in terms of increased computational power; we have merely demonstrated that the NEF is compatible with such neurons.

The biological systems are intrinsically complex, and therefore, it is understandable that models show limitations. The authors don’t need to change their model or methods but should either tone down their claims or justify their methods through discussions.

We agree with your points and have clarified the manuscript as discussed above. That is, we have emphasized the role of the tuning curves used by the NEF and explain our focus on passive neurons.

We have furthermore expanded on the limitations of our approach as pointed out by your comments in the discussion section. The following lines have been added/expanded on in the Discussion (pages 24-25 of the revised manuscript):

It will be important to determine what kinds of single cell dynamics can be successfully incorporated into such models in practice and what limitations there are.  The majority of examples we have explored to date have dynamics that are not as complicated as those found in sophisticated multi-compartmental single cells models. It is generally possible to include more detailed passive dendrite models in the NEF [28]. Additionally, prior work has adapted the NEF to accommodate models with biologically detailed active dendrites and conduction-based synapses, but at the cost of a more computationally expensive training regime [19]. It remains to be seen if the tools presented in this work could be successfully deployed on this more biologically complex version of the NEF. 

3, the link to the code does not work.

The code repository is planned to be made public upon publication. Though it could be opened sooner if requested by the reviewers and/or editors.

Round 2

Reviewer 1 Report

I have reviewed the manuscript and I am satisfied with the responses given to my comments. The authors have addressed all of my concerns and I believe the manuscript is ready for publication. I, therefore, recommend acceptance of the manuscript.

Author Response

Thank you reviewer 1 for all of your feedback.

Reviewer 3 Report

Overall, the authors solved my concerns. However, the authors should cite and discuss relevant papers when discussing complex neuronal dynamics, including the work by Tzilivaki et al., Nat. Comms 2019, Zang et al., eLife 2020 and Zang & De Schutter, J. Neurosci 2021. We all know neuronal dendrites are not passive. Therefore, only citing the paper from their own group regarding passive dendrites may be misleading.

Author Response

Thank you reviewer 3 for the suggested citations. We've reviewed the papers and they provide great examples of the effect of dendrites on neural dynamics. We've incorporated these citations into our discussion section:

"Additionally, active dendrites and conduction-based synapses have been 793
modelled in, what’s called, the oracle-supervised Neural Engineering Framework (osNEF) 794
[19]. While, this greater biological fidelity comes at the cost of a more computationally 795
expensive training regime, this is necessary to model the dynamics of actual neurons. 796
Indeed, active dendrites are necessary to capture the complex neuronal dynamics of, for 797
example, FS basket cells in hippocampus (Tzilivaki 2019) and Purkinje cells of the cerebellar cortex 798
(Zang 2020, Zang 2021). It remains to be seen if the tools presented in this work could be successfully 799
deployed on the more biologically complex osNEF.